# Experimental Evaluation and Modeling of Physical Hardening in Asphalt Binders

**DOI:** 10.3390/ma15010019

**Published:** 2021-12-21

**Authors:** Chiara Tozzi, Davide Dalmazzo, Orazio Baglieri, Ezio Santagata

**Affiliations:** Department of Environment, Land and Infrastructure Engineering, Politecnico di Torino, Corso Duca degli Abruzzi 24, 10129 Torino, Italy; chiara.tozzi@polito.it (C.T.); davide.dalmazzo@polito.it (D.D.); orazio.baglieri@polito.it (O.B.)

**Keywords:** asphalt binder, physical hardening, glass transition, creep, bending beam rheometer, hardening rate

## Abstract

The research described in this paper deals with the experimental evaluation and modeling of physical hardening in asphalt binders. The term physical hardening refers to a reversible phenomenon occurring at low temperatures that causes time-dependent changes in viscoelastic properties. The experimental approach, followed to quantitatively assess physical hardening, was based on flexural creep tests carried out by means of the Bending Beam Rheometer at various temperatures and conditioning times. The results obtained confirmed that hardening phenomena have a significant influence on the creep response of asphalt binders, to an extent that can be quantitatively assessed by referring to the appropriate rheological parameters and by applying the loading time–conditioning time superposition principle. The experimental data were fitted to a mechanical model proposed in the literature (composed of a single Kelvin–Voigt element) and thereafter to an improved model (with two Kelvin–Voigt elements in series). Both models were assessed in terms of their prediction accuracy. The improved model was found to better describe physical hardening effects in the case of both short- and long-term conditioning. Practical implications of the study were finally highlighted by referring to possible ranking criteria to be introduced in acceptance procedures for the comparative evaluation of asphalt binders.

## 1. Introduction

Thermal cracking is one of the main distress types affecting asphalt pavements, typically occurring in regions characterized by cold climatic conditions associated with significant daily temperature variations. As proven by the abundant literature on the subject, it is well recognized that the rheological properties of asphalt binders employed in pavement surface layers play a major role in controlling such distress [1,2,3]. In fact, due to their viscoelastic nature, binders have the capability to relax stresses induced by temperature decrease; however, if these stresses are not dissipated, they can be released by crack formation once the tensile strength of the material has been exceeded [4].

Low-temperature physical hardening is a phenomenon that causes time-dependent isothermal changes in rheological properties of asphalt binders. The hardening mechanism can be explained by the “free volume theory” [5,6]. Such a theory assumes the material’s total volume to be composed of two fractions: the first fraction (occupied volume) is the volume of molecules, including their vibrational motions; the second fraction (free volume) is the volume of voids between molecules, caused by packing irregularities. When an asphalt binder is subjected to cooling starting from high temperatures, it initially experiences configurational rearrangements of its molecules and, as a result, its free volume decreases. As temperature continues to decrease, particle mobility progressively decreases until a certain temperature, at which it becomes so small that the reduction in free volume cannot be accomplished within the experimental time. In such conditions, free volume is greater than the volume corresponding to equilibrium, and the material is therefore in an unstable state. However, if kept at constant temperature, the binder progressively tends toward equilibrium with a gradual volume change referred to as physical hardening. Such a phenomenon has a non-negligible impact on the mechanical response of the material. In particular, it has been shown to cause an increase in the binder’s stiffness and a reduction in its stress relaxation capability [7].

The overall consequence of physical hardening is that it may significantly contribute to the build-up of increased and persistent thermal stresses in asphalt mixtures, leading to the premature failure of pavement structures. Thus, evaluation of physical hardening effects is of paramount importance in the performance characterization of asphalt binders. 

Physical hardening in asphalt binders was first documented in 1977 by Struik [7], who observed a change in the creep response of materials subjected to different isothermal conditioning times. He also observed that creep curves obtained at increasing conditioning times appeared to be shifted along the loading time scale, indicating that all retardation times changed by the same factor. This finding suggested the existence of a loading time–conditioning time relationship which could be exploited for the construction of continuous master curves for creep response functions.

After the pioneering work of Struik, a comprehensive study on physical hardening of asphalt binders was carried out through the Strategic Highway Research Program (SHRP) [8]. Researchers observed the existence of a correspondence between the temperature interval in which hardening effects were significant and the glass transition region [9]. Unlike polymers and other amorphous materials, for which physical hardening was reported to occur below the glass transition temperature, in the case of asphalt binders physical hardening was observed both below and above the glass transition point. This peculiar behavior was attributed to the crystallization of waxes, and it was demonstrated that the rate of hardening was affected not only by the rate of free volume collapse, but also by the relative amount of wax crystalline fractions progressively formed in the asphalt binder [10]. 

In recent years, several investigations have highlighted the impact of physical hardening on low-temperature pavement cracking [11,12] and identified the need to improve asphalt binder specifications to take this phenomenon into account [13,14,15].

Tabatabaee et al. [16] showed that physical hardening can be quantitively described as a function of isothermal conditioning time by means of a modified version of the classical Kelvin–Voigt mechanical creep model, in which strain variations are replaced by relative changes in volume, which are in turn considered to be proportional to relative changes in stiffness. Based on the results obtained from an experimental study that involved flexural creep tests conducted in different testing conditions, the authors showed that the amount of physical hardening accounted for by the model could be expressed as a function of glass transition temperature, conditioning time, and test temperature. In such a context, they also underlined that model variables are unique material parameters that remain constants in all testing conditions.

The modeling approach proposed by Tabatabaee et al. [16] relies on the use of a single Kelvin–Voigt element that entails the existence of a single retardation time for any combination of test temperature and conditioning time. However, the use of a single time factor may be not sufficient to adequately describe the evolution due to hardening, especially in the case of long conditioning times imposed to test specimens. Moving from this observation and based on experimental results, gathered from laboratory tests conducted on different asphalt binders, the research described in this paper aimed at verifying the effectiveness of the above-mentioned model in predicting physical hardening under long-term conditioning times. Following such an assessment, a modified model was proposed and was thereafter found to provide a better description of the short- and long-term hardening behavior of the considered binders. 

## 2. Materials and Methods

Two different asphalt binders were used in the experimental study. They were provided by refineries which operate on crudes of various origins, and they differed in terms of penetration grade as follows: 40/60 (binder A) and 70/100 (binder B).

With the purpose of highlighting the effect of physicochemical properties on physical hardening phenomena, the asphalt binders were subjected to a preliminary characterization that included fractionation analysis (also referred to as SARA analysis) and determination of glass transition temperature (T_g_).

Fractionation analysis was carried out by making use of a methodology that combines thin layer chromatography (TLC, LSI Medience Corporation, Tokyo, Japan) and flame ionization detection (FID, LSI Medience Corporation, Tokyo, Japan) [17,18,19]. TLC allows the separation of SARA fractions (saturates, aromatics, resins and asphaltenes) through their successive elution in solvents of increasing polarity, while FID allows for the quantification of the relative amounts of the four fractions.

The results obtained from the analysis of the two binders, performed with a Iatroscan MK-6 analyzer (LSI Medience Corporation, Tokyo, Japan), are reported in Table 1 along with the values of the Gaestel index (I_C_), which were calculated as the ratio of the cumulative percentages of asphaltenes and saturates to the sum of the percentages of aromatics and resins [20]. The experimental data indicate that the two binders exhibited a substantial similarity in terms of saturates and aromatics, while non-negligible differences were recorded for the other two fractions (resins and asphaltenes). In particular, binder A was found to contain a lower percentage of resins and a higher percentage of asphaltenes with respect to binder B. Overall, composition data led to a higher Ic value for binder A with respect to binder B, indicating the former to be characterized by a denser gel-like colloidal structure [21].

Determination of T_g_ involved the use of a dynamic shear rheometer (DSR, Anton Paar, Graz, Austria) equipped with 4 mm parallel plates. The test procedure employed, previously proposed by Santagata et al. [22], requires cylindrical stress-free specimens to be subjected to a uniform cooling rate of 1 °C/min, with the corresponding progressive monitoring of vertical gap reduction. By neglecting transverse deformation of the specimen, such a measurement is assumed to be representative of volume reduction and is therefore employed for the determination of T_g_. In particular, as displayed in Figure 1, the value of T_g_ is derived from the temperature–gap reduction plot by considering the interception point between the thermo-dynamic equilibrium line (identified at higher temperatures) and the glassy non-equilibrium line (associated with lower temperatures). The experimental results, obtained as the average of two replicates, highlighted the existence of a slight difference between the two considered binders, with binder A characterized by a higher T_g_ value with respect to binder B (−21.8 °C vs. −23.1 °C).

Rheological effects caused by physical hardening phenomena were assessed by means of low-temperature creep tests conducted with a bending beam rheometer (BBR, Cannon Instrument Company, State College, Pennsylvania) as per AASHTO T 313. The rheological parameters considered in the analysis were creep stiffness (S) and creep rate (m). As shown in the testing program provided in Table 2, tests were carried out at different temperatures (T). The specimen for each test temperature was conditioned isothermally at the test temperature considering different conditioning times (t_c_). All measurements were performed in at least two replicates, and the average results were thereafter considered in the analysis. The acceptance range of two test results (d2s) was set equal to 3% for both S and m.

## 3. Experimental Results

Figure 2 shows an example of results obtained from BBR tests, represented in the form of creep stiffness vs. loading time plots at various conditioning times (for binder A at −24 °C). In accordance with expectations, with the increase in conditioning time the creep stiffness data points were observed to be progressively shifted towards the upper part of the diagram as a consequence of physical hardening. The magnitude of such a shift (i.e., magnitude of hardening) was found to be binder-specific and dependent, for each material, upon test temperature.

By considering the standard SHRP low-temperature parameter S_60_ (i.e., creep stiffness at 60 s loading time), Tabatabaee et al. [16] formulated and validated their model by referring to the relative increase in creep stiffness due to physical hardening, indicated as “hardening rate” (HR). Such a parameter can therefore be computed, as shown in the following equation:(1)$$\mathrm{HR}=\frac{S_{60,i}-S_{60,r}}{S_{60,r}}$$
where S_60,i_ and S_60,r_ are the values of S_60_ corresponding to a given conditioning time t_ci_ and a reference conditioning time t_cr_, respectively, with t_cr_ assumed to be equal to 1 h.

The values of HR plotted as a function of conditioning time, for the two binders considered in the investigation (A and B), are reported in Figure 3. It can be observed that the values of HR for binder A increased when passing from −12 °C to −18 °C at all conditioning times, while they significantly decreased when passing from −18 °C to −24 °C. A slightly different outcome was recorded for binder B, which exhibited similar HR values at −12 °C and −18 °C, higher than those recorded at −24 °C. These results collectively suggest the existence of a characteristic temperature at which physical hardening effects can reach a peak, hereafter indicated as “hardening temperature” (T_H_). The occurrence of such a peak in the case of the considered binders can be better appreciated by considering Figure 4, which was obtained by plotting the same HR values shown in Figure 3 as a function of temperature, for various conditioning times. Although the exact position of the HR peaks cannot be precisely identified due to the limited number of test temperatures considered in the investigation, the data displayed indicate that T_H_ is close to −18 °C for binder A and in the range from −12 °C to −18 °C for binder B. Available data do not exclude the possibility that such a temperature may depend upon conditioning time.

The experimental results illustrated above are in good agreement with the discussion provided by Tabatabaee et al. [16]. However, while these researchers could assume peak hardening temperature to be equal to T_g_, in the investigation described in this paper T_H_ values were found to differ significantly from the T_g_ values reported in Section 2. Such a difference may be due to the cooling rate occurring in BBR tests, which is definitely higher than that imposed by the DSR during the procedure employed for the measurement of T_g_.

An alternative analysis of the experimental data presented above can be carried out by applying the loading time–conditioning time superposition principle. At each temperature, values of a selected rheological function (e.g., creep stiffness) are plotted as a function of loading time for each considered conditioning time. After selecting a reference conditioning time (e.g., 1 h), the sets of data points corresponding to the considered conditioning times are shifted in the direction of the horizontal loading time axis, thereby resulting in a smooth and continuous master curve. Shift factors (a_tc_) that are necessary in order to achieve such a result may be considered as intrinsic properties of each considered binder, revealing its sensitivity to physical hardening.

Master curves obtained at various temperatures for the two binders subjected to testing are shown in Figure 5, while the corresponding shift factors are presented in Figure 6. For both binders it was observed that shift factors increased with the decrease in temperature, thereby showing a trend different from that exhibited by HR (Figure 3). Nevertheless, as shown in Figure 7, a good correlation was found between HR and log(a_tc_), with R^2^ values greater than 0.96 for all considered temperatures.

## 4. Hardening Rate Modeling

The HR values determined from the experimental data for both considered binders (A and B) were fitted to the model proposed by Tabatabaee et al. [16], formally expressed by the following equations:(2)$$\mathrm{HR}\left(T,t_{c}\right)\;=\;\frac{\varphi \left(T\right)\;}{G}\left(1-e^{-t_{c}\frac{G}{\eta }}\right)$$
(3)$$\varphi \left(T\right)\;=\;e^{\frac{-9{\left(T-T_{0}\right)}^{2}}{{\left(2x\right)}^{2}}}$$
in which G and η are viscoelastic parameters of the mechanical creep model, T_0_ is the temperature which corresponds to peak hardening rate, and 2x is the length of the temperature range in which hardening occurs.

Since it was observed that the measured T_g_ values of the considered binders differed significantly from the T_H_ values estimated from T-HR plots (Figure 4), while fitting experimental data to the model T_0_ was not fixed equal to T_g_, as hypothesized by Tabatabaee et al. [16]. Rather, T_0_ was considered as one of the four model parameters to be calculated by minimizing deviations between measured and calculated HR values. 

The outcomes of the fitting process described above are reported in Table 3. It can be observed that, for both binders, calculated values of T_0_ were found to be significantly higher than T_g_ values and to be in good agreement with the estimates derived from the abovementioned T-HR plots. Furthermore, as revealed by the calculated values of 2x, binder B appeared to be characterized by a wider hardening zone compared to binder A. 

A comparison between HR values determined from experimental data and from model fitting is provided in Figure 8 (binder A) and in Figure 9 (binder B). The 1-element model seems to adequately represent the time dependency of HR for binder B (with an overall goodness of fit R^2^ equal to 0.97), whereas it underestimates the long-term hardening effects for binder A (with a consequent lower value of R^2^, equal to 0.93). From the model curves and data points shown in Figure 8 it can also be observed that, for binder A, the horizontal asymptote of HR is reached too soon, while this occurs at higher conditioning times and in better agreement with experimental data in the case of binder B (Figure 9). Such an outcome is consistent with the calculated values of retardation time (η/G), equal to 22.48 h in the case of binder B and 6.31 h for binder A.

With the purpose of improving the prediction accuracy of the model and of achieving a better fit with experimental data, a modified version was considered in the analysis. Such a version of the model is composed of two Kelvin–Voigt elements in series, and the corresponding equation for the calculation of HR as a function of T and t_c_ can be written as follows:(4)$$\mathrm{HR}\left(T,t_{c}\right)\;=\;\varphi \left(T\right)\;\cdot \;\left[\frac{\left(1-e^{-t_{c}\frac{G_{1}}{\eta_{1}}}\right)}{G_{1}}+\frac{\left(1-e^{-t_{c}\frac{G_{2}}{\eta_{2}}}\right)}{G_{2}}\right]$$
in which G_1_, η_1_, G_2_, and η_2_ are viscoelastic parameters of the mechanical creep model, while φ(T) is given by Equation (3).

The results obtained by fitting the modified model to the experimental data are presented in Figure 10 (for binder A) and Figure 11 (for binder B), while the values of model parameters are given in Table 4. As expected, the presence of the second Kelvin–Voigt element improved the goodness of fit with respect to the 1-element model. Such an improvement was more significant in the case of binder A, for which the evolution of HR is adequately described along the entire scale of conditioning times, as proven by the very high value of R^2^ (equal to 0.99). Nevertheless, a slight improvement was also recorded for binder B (with R^2^ changing from 0.97 to 0.98). It is interesting to observe that, for both binders, the calculated values of T_0_ changed only slightly with respect to those obtained from the 1-element model. A significant change of 2x was recorded only in the case of binder A, with the identification of a narrower hardening range. However, a more realistic and gradual description of physical hardening effects was obtained since the proposed model entails the existence of two retardation times (η_1_/G_1_ and η_2_/G_2_). The first and shorter retardation time (η_1_/G_1_) affects the shape of the HR function in the short term, while the second and longer retardation time (η_2_/G_2_) mainly controls the long-term response.

By making use of both considered models (composed of either one or two Kelvin–Voigt elements), the maximum value of HR (indicated as HR_max_) that can be achieved in the long term as a result of physical hardening can be computed at any test temperature. This is shown in Figure 12, where the displayed curves are represented for each binder in the hardening region with an extension of 2x centered on T_0_. For both binders, the 2-element model leads to higher and more realistic peak values of HR_max_.

## 5. Conclusions 

Based on the results discussed in this paper, it may be concluded that low-temperature physical hardening phenomena occurring in asphalt binders can be comprehensively described by performing BBR creep tests at different temperatures and conditioning times, and by thereafter fitting experimental results to the improved mechanical model composed of two Kelvin–Voigt elements combined in series. Such an approach allows the rheological effects of physical hardening to be quantitatively assessed at any temperature and conditioning time in terms of the expected relative increase in creep stiffness (HR). Furthermore, use of the model can lead to the construction of binder-specific characteristic curves in which the maximum calculated value of relative stiffness change (HR_max_) is plotted as a function of temperature. 

Although the investigation was limited to only two binders of the neat type (differing in penetration grade), in principle the proposed approach can be employed for the comparative evaluation of binders of different types, including polymer-modified ones. Corresponding rankings can be identified based on the HR values obtained in specific temperature–conditioning time combinations or with respect to the maximum stiffening potential quantified at any given temperature in terms of the corresponding value of HR_max_.

In order to validate the proposed approach and introduce it in specification frameworks, additional investigations are needed. In particular, further efforts are required in defining criteria for the selection of test conditions that need to be considered to obtain a meaningful and reliable fitting of the model to experimental data. Furthermore, improvements may be sought in the assessment of the so-called hardening temperature (T_H_) since it would be convenient to derive it from glass transition temperature (T_g_), thereby eliminating one of the six model parameters to be found by means of model fitting.

## Figures and Tables

**Figure 1 materials-15-00019-f001:**
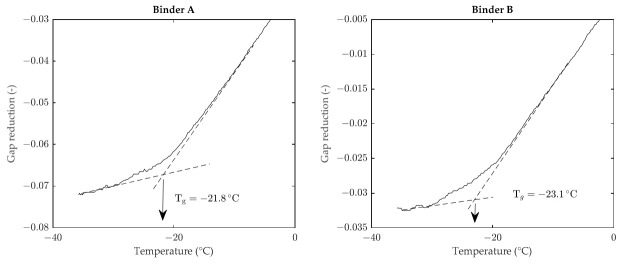
Determination of the glass transition temperature.

**Figure 2 materials-15-00019-f002:**
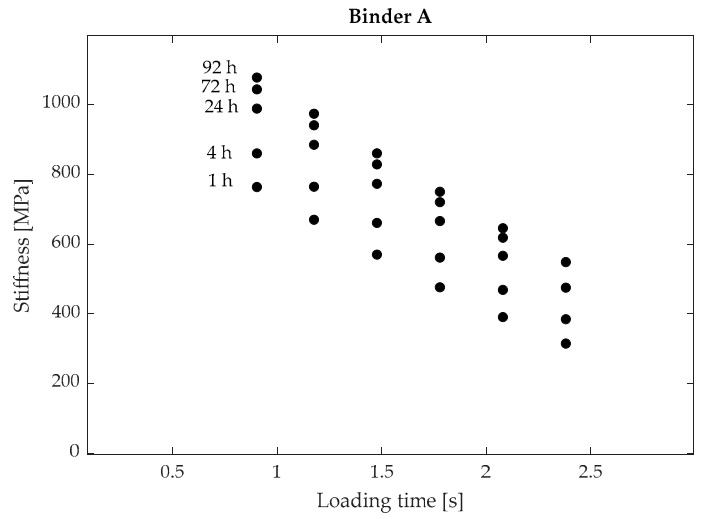
Creep stiffness as a function of loading time at different conditioning times (binder A at −24 °C).

**Figure 3 materials-15-00019-f003:**
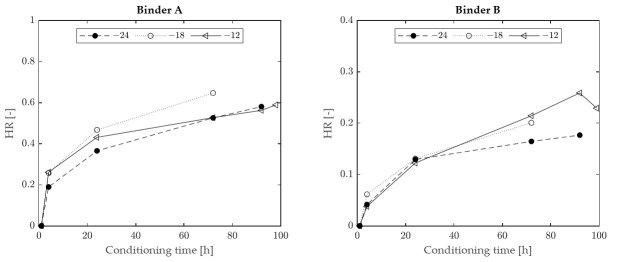
Hardening rate as a function of conditioning time (experimental data).

**Figure 4 materials-15-00019-f004:**
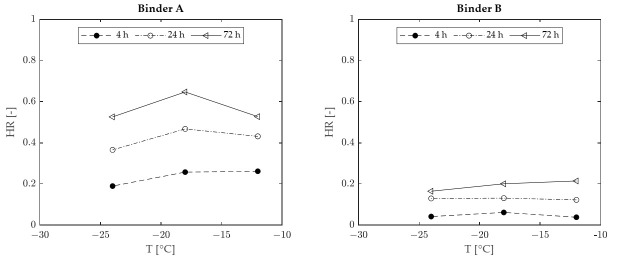
Hardening rate as a function of temperature (experimental data).

**Figure 5 materials-15-00019-f005:**
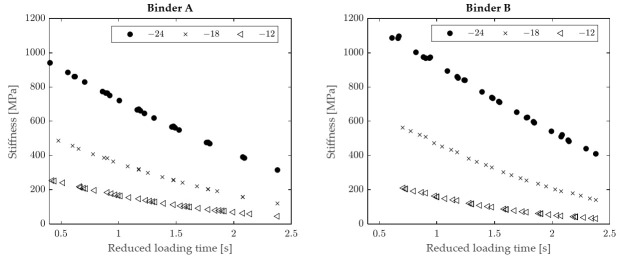
Experimental stiffness at conditioning aging time of 1 h, for different temperatures, as a function of loading time.

**Figure 6 materials-15-00019-f006:**
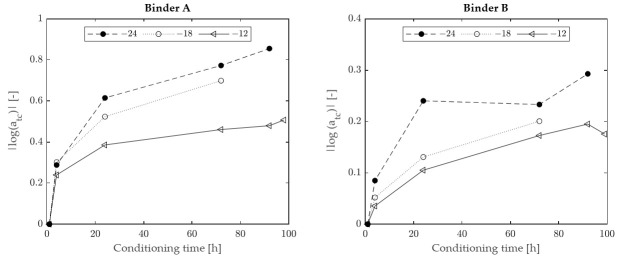
Horizontal shift factors, at different temperatures, as a function of conditioning time.

**Figure 7 materials-15-00019-f007:**
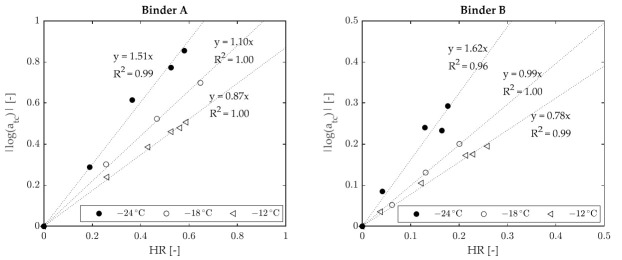
Relationship between horizontal shift factors and HR.

**Figure 8 materials-15-00019-f008:**
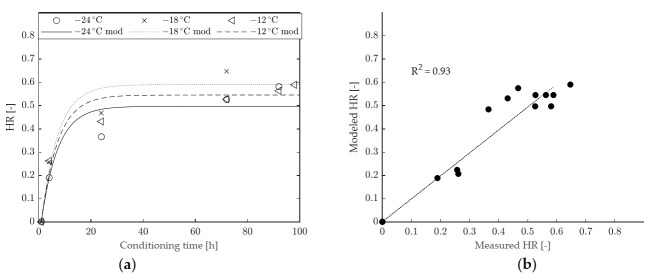
HR of binder A: (**a**) comparison between modeled and experimental data, (**b**) goodness of fit of the 1-element model.

**Figure 9 materials-15-00019-f009:**
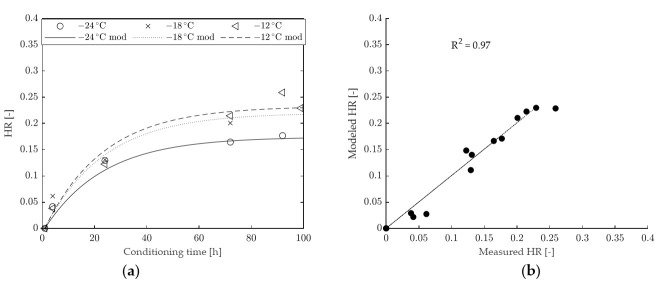
HR of binder B: (**a**) comparison between modeled and experimental data, (**b**) goodness of fit of the 1-element model.

**Figure 10 materials-15-00019-f010:**
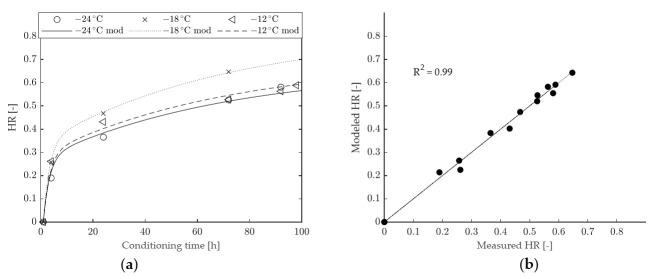
HR of binder A: (**a**) comparison between modeled and experimental data, (**b**) goodness of fit of the improved model.

**Figure 11 materials-15-00019-f011:**
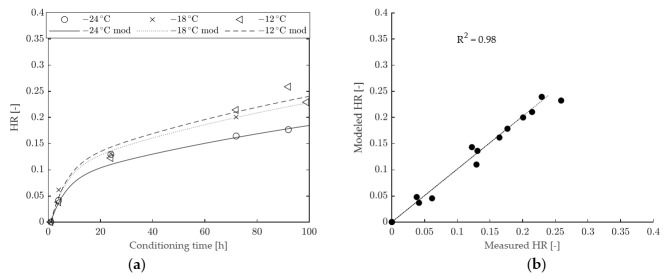
HR of binder B: (**a**) comparison between modeled and experimental data, (**b**) goodness of fit of the improved model.

**Figure 12 materials-15-00019-f012:**
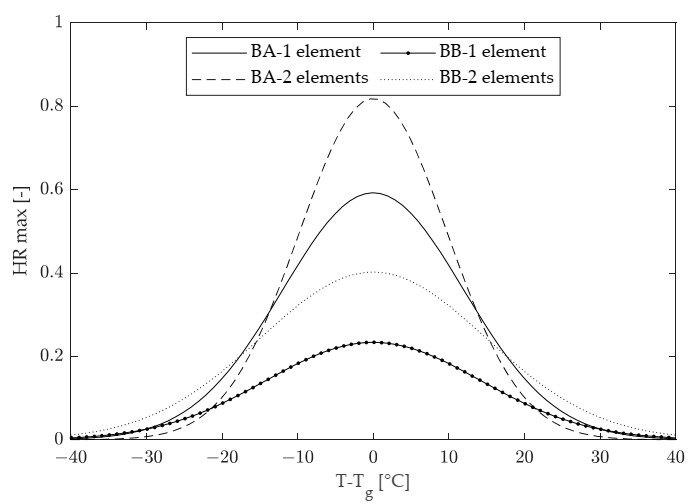
Maximum values of the modeled HR as a function of temperature.

**Table 1 materials-15-00019-t001:** SARA fractions and Gaestel index of asphalt binders.

Binder	Saturates [%]	Aromatics [%]	Resins [%]	Asphaltenes [%]	$I_{C}\;[-]$
A (40/60)	5.1	45.4	18.7	30.8	0.56
B (70/100)	6.1	44.1	22.8	26.9	0.49

**Table 2 materials-15-00019-t002:** Test temperatures and conditioning times adopted in BBR tests.

T [°C]	t_c_ [h]
−12	1, 4, 24, 72, 92, 98
−18	1, 4, 24, 72
−24	1, 4, 24, 72, 92

**Table 3 materials-15-00019-t003:** One-element model parameters.

Model Parameters	Binder A	Binder B
$T_{0}$ [°C]	−16.9	−12.4
2x [°C]	50.8	64.15
G [MPa]	1.69	4.28
η [MPa∙h]	10.66	96.24

**Table 4 materials-15-00019-t004:** Improved model parameters.

Model Parameters	Binder A	Binder B
$T_{0}$ [°C]	−17.6	−13.2
2x [°C]	41.5	63.0
G_1_ [MPa]	2.92	9.31
η_1_ [MPa∙h]	6.96	55.35
G_2_ [MPa]	2.10	3.33
η_2_ [MPa∙h]	148.97	558.09

## Data Availability

The data presented in this study are available on request from the corresponding author.

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
