# Peer review of "Experimental Evaluation and Modeling of Physical Hardening in Asphalt Binders"

_materials, 2021, doi:10.3390/ma15010019_

Round 1

Reviewer 1 Report

Very interesting paper. Physical hardening or thermoreversible aging is an important factor that leading to premature cracking of asphalt pavement. The authors give a good experimental design on this topic. However, the authors are suggested to further modify and improve the following issues.

  1. The authors’ references are mostly early studies on physical hardening in asphalt binder. The authors ignore the recent contributions of some scholars to the research on asphalt physical hardening, such as the work of Professor Simon Hesp's research group (Hesp et al. and Ding et al.).
  2. Currently, the degree of physical hardening in asphalt binder is usually quantified by the Grade Loss (GL) obtained from the extended bending beam rheometer (ExBBR) test (AASHTO TP T122 method). The authors used “hardening rate” (HR) in the manuscript. However, this index cannot fully reflect the adverse effects of physical hardening on the performance grading of asphalt binder, such as relaxation properties.
  3. The citation format needs to be revised (Line 164; Line 172; Line 173; Line 199; Line 200; Line 257…).
  4. Avrami and Ozawa theory are also used by other researchers to model the isothermal and non-isothermal hardening process. The authors are recommended to further fit the data with crystallization kinetic theory.

Author Response

Reviewer #1

Comment

Reply

#1

Very interesting paper. Physical hardening or thermoreversible aging is an important factor that leading to premature cracking of asphalt pavement. The authors give a good experimental design on this topic. However, the authors are suggested to further modify and improve the following issues.

The Authors wish to thank Reviewer #1 for the positive comment on their work.

#2

The authors’ references are mostly early studies on physical hardening in asphalt binder. The authors ignore the recent contributions of some scholars to the research on asphalt physical hardening, such as the work of Professor Simon Hesp's research group (Hesp et al. and Ding et al.).

Done

The Authors selected the references with respect to the primary purpose of the study, aimed to model the physical hardening phenomenon in time, and to predict long-term hardening using relatively short-term laboratory testing.

In this respect, a complete review on the subject and of current practices was out of the scope. Nevertheless, the Authors agree with the Reviewer’s comment, and some recent contributions were added in the amended version.

The following part was added in the Introduction:

“In recent years, several investigations highlighted the impact of physical hardening on low temperature pavement cracking [11,12] and identified the need to improve as-phalt binder specifications to take this phenomenon into account [13, 14, 15].”

The following references were added to the manuscript:

11.        Hesp, S.A.M.; Soleimani, A.; Subramani, S.; Phillips, T.; Smith, D.; Marks, P.; Tam, K.K. Asphalt Pavement Cracking: Analysis of Extraordinary Life Cycle Variability in Eastern and Northeastern Ontario. Int J Pavement Eng 2009, 10, 209–227, doi:10.1080/10298430802343169.

12.        Shurvell, H.; Subramani, S.; Hesp, S. a. M.; Genin, S.; Scafe, D. Five Year Performance Review of a Northern Ontario Pavement Trial: Validation of Ontario’s Double-Edge-Notched Tension (DENT) and Extended Bending Beam Rheometer (BRR) Test Methods.; In: Fifty-Fourth Conference of the Canadian Technical Asphalt Association; 2009.

13.        Berkowitz, M.; Filipovich, M.; Baldi, A.; Hesp, S.A.M.; Aguiar-Moya, J.P.; Lorı́a-Salazar, L.G. Oxidative and Thermoreversible Aging Effects on Performance-Based Rheological Properties of Six Latin American Asphalt Binders. Energy Fuels 2019, 33, 2604–2613, doi:10.1021/acs.energyfuels.8b03265.

14.        Ding, H.; Tetteh, N.; Hesp, S.A.M. Preliminary Experience with Improved Asphalt Cement Specifications in the City of King-ston, Ontario, Canada. Constr Build Mater 2017, 157, 467–475, doi:10.1016/j.conbuildmat.2017.09.118.

15.        Li, Y.; Ding, H.; Nie, Y.; Hesp, S.A.M. Effective Control of Flexible Asphalt Pavement Cracking through Quality Assurance Testing of Extracted and Recovered Binders. Constr Build Mater 2021, 273, 121769, doi:10.1016/j.conbuildmat.2020.121769.

#3

Currently, the degree of physical hardening in asphalt binder is usually quantified by the Grade Loss (GL) obtained from the extended bending beam rheometer (ExBBR) test (AASHTO TP T122 method). The authors used “hardening rate” (HR) in the manuscript. However, this index cannot fully reflect the adverse effects of physical hardening on the performance grading of asphalt binder, such as relaxation properties.

As stated in the previous comment, the study aimed to model the physical hardening phenomenon in time and not to investigate detrimental effects of physical hardening on performance ranking. For this reason, the Performance Grade was not discussed in the article.

The HR index was used for the purpose of following (and improving) the model proposed and successfully used by H.A. Tabatabaee, R. Velasquez and H.U. Bahia (10.1016/j.conbuildmat.2012.02.039). The model is relatively new and has never been used or discussed in literature. Thus, it needs additional experimental investigations to be validated and/or improved.

#4

The citation format needs to be revised (Line 164; Line 172; Line 173; Line 199; Line 200; Line 257…).

Done

#5

Avrami and Ozawa theory are also used by other researchers to model the isothermal and non-isothermal hardening process. The authors are recommended to further fit the data with crystallization kinetic theory.

The Authors thank the Reviewer for the suggestion. This study focused on the validation and discussion of the Kelvin-Voigt model applied to the description of physical hardening. Nevertheless, the Authors will take the Reviewer’s suggestion into consideration in further investigations on the topic.

Reviewer 2 Report

Accepted with major modification.

The topic of the paper is important and new. The manuscript is needed to be revised. However, some comments need to be addressed:

  • The Literature review is too short, and would expect much on the technical issue that reflects the evaluation and modelling of physical hardening in asphalt binders of the manuscript?
  • Authors can cite the following work in the literature review section, which is closely related to their work, and some are recently reported?
  1. https://doi.org/10.1016/j.conbuildmat.2016.10.073,
  2. https://doi.org/10.1016/j.conbuildmat.2021.124687,
  3. https://doi.org/10.3390/app11146521
  • What makes this paper different from the other and the oldest ones?

  • The authors pointed out the current study, which is implemented evaluation and modelling of physical hardening in asphalt binders but did not give possible directions for future studies?
  •  Error! Reference source not found in line 164 and others ?

Author Response

Reviewer #2

Comment

Reply

#1

The topic of the paper is important and new. The manuscript is needed to be revised. However, some comments need to be addressed:

#2

The Literature review is too short, and would expect much on the technical issue that reflects the evaluation and modelling of physical hardening in asphalt binders of the manuscript?

Authors can cite the following work in the literature review section, which is closely related to their work, and some are recently reported?

https://doi.org/10.1016/j.conbuildmat.2016.10.073,

https://doi.org/10.1016/j.conbuildmat.2021.124687,

https://doi.org/10.3390/app11146521

Done

The Authors selected the references with respect to the primary purpose of the study, aimed to model the physical hardening phenomenon in time, and to predict long-term hardening using relatively short-term laboratory testing.

In this respect, a complete review on the subject and of current practices was out of the scope. Nevertheless, the Authors agree with the Reviewer’s comment, and some recent contributions were added in the amended version.

The following part was added in the Introduction:

“In recent years, several investigations highlighted the impact of physical hardening on low temperature pavement cracking [11,12] and identified the need to improve as-phalt binder specifications to take this phenomenon into account [13, 14, 15].”

The following references were added to the manuscript:

11.        Hesp, S.A.M.; Soleimani, A.; Subramani, S.; Phillips, T.; Smith, D.; Marks, P.; Tam, K.K. Asphalt Pavement Cracking: Analysis of Extraordinary Life Cycle Variability in Eastern and Northeastern Ontario. Int J Pavement Eng 2009, 10, 209–227, doi:10.1080/10298430802343169.

12.        Shurvell, H.; Subramani, S.; Hesp, S. a. M.; Genin, S.; Scafe, D. Five Year Performance Review of a Northern Ontario Pavement Trial: Validation of Ontario’s Double-Edge-Notched Tension (DENT) and Extended Bending Beam Rheometer (BRR) Test Methods.; In: Fifty-Fourth Conference of the Canadian Technical Asphalt Association; 2009.

13.        Berkowitz, M.; Filipovich, M.; Baldi, A.; Hesp, S.A.M.; Aguiar-Moya, J.P.; Lorı́a-Salazar, L.G. Oxidative and Thermoreversible Aging Effects on Performance-Based Rheological Properties of Six Latin American Asphalt Binders. Energy Fuels 2019, 33, 2604–2613, doi:10.1021/acs.energyfuels.8b03265.

14.        Ding, H.; Tetteh, N.; Hesp, S.A.M. Preliminary Experience with Improved Asphalt Cement Specifications in the City of King-ston, Ontario, Canada. Constr Build Mater 2017, 157, 467–475, doi:10.1016/j.conbuildmat.2017.09.118.

15.        Li, Y.; Ding, H.; Nie, Y.; Hesp, S.A.M. Effective Control of Flexible Asphalt Pavement Cracking through Quality Assurance Testing of Extracted and Recovered Binders. Constr Build Mater 2021, 273, 121769, doi:10.1016/j.conbuildmat.2020.121769.

#3

What makes this paper different from the other and the oldest ones?

The investigation carried out in this paper aimed to evaluate, discuss and improve the model proposed by H.A. Tabatabaee, R. Velasquez and H.U. Bahia (10.1016/j.conbuildmat.2012.02.039). The model represents a powerful tool for the laboratory characterization of asphalt binders, since it allows to predict the long-term hardening through relatively short-term testing. The model is relatively new and not used or discussed in literature and therefore needs several experimental investigations to be validated. Furthermore, the present study proposes some modifications to the model to ensure that both the short- and long-term ageing can be predicted.

#4

The authors pointed out the current study, which is implemented evaluation and modelling of physical hardening in asphalt binders but did not give possible directions for future studies?

The Authors gave some possible directions for further studies in the conclusion of the paper. In particular, the following part can be underlined:

“In order to validate the proposed approach and to introduce it in specification frameworks, additional investigations are needed. In particular, further efforts are required in defining criteria for the selection of test conditions that need to be considered to obtain a meaningful and reliable fitting of the model to experimental data. Furthermore, improvements may be sought in the assessment of the so-called hardening temperature (TH) since it would be convenient to derive it from glass transition temperature (Tg), thereby eliminating one of the six model parameters to be found by means of model fitting.”

#5

Error! Reference source not found in line 164 and others?

Done

The references were correctly formatted.

Reviewer 3 Report

Thank you very much for the interesting contribution. I have only very few comments and proposals for additions, which could help the reader to follow the discusions. The proposals are optional. 

  • Introduction: A figure explaining the physical hardening by free voulme theory would help the understanding of the theory
  • Materials & Methods: Please mention the conditioning parameters (temperature). Is it the same as the test temperature? This is not precisely defined in the paper and should be added in the text and table 1 caption.
  • A equation for shift factors or their display in one figure (fig 5?) would help to follow the concept
  • Please check figure numbers and references

Author Response

Reviewer #3

Comment

Reply

#1

Thank you very much for the interesting contribution. I have only very few comments and proposals for additions, which could help the reader to follow the discussions. The proposals are optional.

The Authors wish to thank Reviewer #3 for the positive comment on their work.

#2

Introduction: A figure explaining the physical hardening by free volume theory would help the understanding of the theory

The Authors provided a brief description of the molecular mechanisms involved in physical hardening. The corresponding figure may be quite complex and in the Authors’ opinion may not be helpful for unexperienced readers. Nevertheless, they will take this suggestion into account in future papers focused on “material science” rather than on modelling.

#3

Materials & Methods: Please mention the conditioning parameters (temperature). Is it the same as the test temperature? This is not precisely defined in the paper and should be added in the text and table 1 caption.

Done

The conditioning temperature is the same temperature referred to as “test temperature”. It was specified in the text as follows:

“The specimen, for each test temperature, was conditioned isothermally at the test temperature, considering different conditioning times (tc)” (line 142).

#4

An equation for shift factors or their display in one figure (fig 5?) would help to follow the concept

The shift factors are displayed in “Figure 6. Horizontal shift factors, at different temperatures, as a function of conditioning time.”

#5

Please check figure numbers and references

Done

Round 2

Reviewer 2 Report

accepted in current shape !